# Urban Sustainable Development Empowered by Cultural and Tourism Industries: Using Zhenjiang as an Example

**Run Liu** [1] and **Ziyue Qiu** [2,*]

1   School of Management, Jiangsu University, Zhenjiang 212013, China
2   College of Environmental & Resource Sciences, Zhejiang University, Hangzhou 310058, China
*   Correspondence: zyqiu@zju.edu.cn; Tel.: +86-571-8898-2615

**Abstract:** Mitigating global warming is a grand challenge for sustainable development in the increasingly urbanized world. How to build a low-carbon society and achieve economic growth at the same time remains less clear. In this paper, using Zhenjiang City in East China as a case study, we analyze the contribution of cultural and tourism industries (CTI) and important low-carbon industries to the sustainable development of the metropolitan area. We found that the CTI in Zhenjiang has accounted for 25% of its total gross domestic product (GDP); the forest recovery for the development of CTI sequestrates 150,000 Mt of carbon dioxide annually, which substantially decreases its carbon emission per GDP and promotes the development of a low-carbon city. With the development of CTI and the transformation of the traditional industrial structure, the tertiary industry has gradually emerged and expanded. CTI-related employment has also increased, contributing to poverty eradication and the achievement of global sustainable development goals. The low-carbon and sustainable development model in Zhenjiang will provide a successful example for other cities, not only within China, but also beyond.

**Keywords:** urban sustainable development; cultural and tourism industry; low-carbon city





## 1. Introduction

Urbanization poses a great threat to global warming due to the huge amount of greenhouse gas (GHG) emission in cities [1]. To achieve urban sustainable development, it is important to mitigate GHG emission in cities. The International Energy Council found that cities account for 70% of global energy consumption and 80% of carbon dioxide ($CO_2$) emissions. A global consensus on climate change and a binding inter-governmental agreement has been reached to reduce $CO_2$ emissions. At the Climate Summit in December 2020, China set a new target to reduce its carbon emissions, whereby $CO_2$ emissions per GDP would fall by > 65% compared to the 2005 level, and non-fossil energy would account for approximately 25% of primary energy consumption. This is the first specific carbon reduction target that China has proposed with the goal of "peaking $CO_2$ emission by 2030 and achieving carbon neutrality by 2060". However, to achieve this goal, urban carbon emissions must be reduced. Furthermore, how to achieve a low-carbon society while ensuring steady economic development is a grand challenge.

Tourism and the environment are a hot issue in urban sustainable development [2]. With socioeconomic development and the improvement of living standards, urban residents are increasingly eager to seek a work–life balance, and to travel in their leisure time. According to the Ministry of Culture and Tourism of China, tourism contributed 11 trillion Yuan to China's gross domestic product (GDP) in 2019, accounting for 11% of the total GDP [3]. In addition, China's tourism industry has created approximately 80 million jobs. Presently, tourism has become one of the pillar industries in China's economic development. However, with the rapid development of tourism, some places fall into the dilemma of destroying the ecological balance due to the blind pursuit of economic benefits [4]. Tourism

and its related transportation industry have a negative impact on the environment and climate change [5].

The development of the tourism industry may inevitably cause pressure on the environment, while the environment is the important prerequisite for tourism development. Thus, the destruction of the environment will reduce the attraction of tourism, and the limitation on the tourism industry will also play a certain role in slowing down the pace of regional economic development [6].

The cultural and tourism industry (CTI) is a significant component of a low-carbon city. Therefore, how to realize the coordination and unity of tourism, economy, and environment is a challenging key for urban sustainable development. Here, we use Zhenjiang City from East China as a case to analyze CTI's contribution to economic growth and low-carbon society construction, aiming at exploring a road of promoting urban sustainable development with a cultural and tourism industry.

## 2. Literature Review

### 2.1. Sustainable Development

Sustainable development was defined in the 1987 report "Our Common Future" from the World Commission on Environment and Development as "development that meets the needs of the present generation without jeopardizing the needs of Future generations on the basis of coordinated and common development of society, economy, population, resources and environment" [7]. Sustainable development, by definition, consists of two interrelated components: "ecological sustainability" and "economic development". Therefore, sustainable development should not only realize the goal of economic development, but also achieve a harmony between natural resources and the environment.

The authority of the definition of sustainable development has been recognized by the academic community and reached consensus in the 1992 United Nations Conference on Environment and Development (UNCED) [8]. Therefore, scholars have studied the theory of sustainable development from multiple disciplines and aspects. At the beginning of the 21st century, the United Nations put forward the Millennium Development Goals. Through the concerted efforts of governments and social organizations over the past decade, remarkable results have been achieved in poverty reduction, education, social justice, and disease prevention. In order to continue meeting the Millennium Development Goals and solve the series of problems facing the world today, the United Nations passed a resolution called Transforming our World: the 2030 Agenda for Sustainable Development in 2015, and paid more attention to the coordination and sustainable development of the global economy, society, and environment, which means that global sustainable development has entered a brand new mechanism framework [9].

### 2.2. Research on the Reform of Industrial Structure Diversification

The diversified development of the urban industrial structure is the important guarantee for urban sustainable development. Many scholars have carried out a great deal of analysis and research in this field. Nowadays, there are many resource-based cities in China, but the development status, regional distribution, and resource reserves differ among these cities, as do the corresponding research methods. Liu [10], Zhang and Wu [11], Yang [12], and many other scholars have conducted detailed and comprehensive research in this area. They analyzed the principle, path, and necessity of industrial structure optimization, and realized that the optimization and reasonable adjustment of industrial structure is the only way to achieve sustainable development of resource-based cities. Wu [13,14] and Yu [15] pointed out the diversification of industrial structure, that is, the deep processing of resources with the support of scientific and technological progress, and the re-selection of leading industries for urban development. They highlighted the overall benefits of the tertiary industry in promoting employment and economic development, and considered it an effective way to optimize the industrial structure.

### 2.3. The Value of the Cultural Tourism Industry to Urban Sustainable Development

According to the UN's 17 Sustainable Development Goals, extreme poverty is to be eradicated for all people around the world by 2030. Poverty eradication is an urgent issue to be addressed in the global development process. At present, theories and practices of poverty alleviation related to industries, society, and tourism have been widely recognized worldwide [16–18]. The research shows that the development of tourism in China has a significant positive effect on economic growth [19]. As the basic strategy for targeted poverty alleviation in rural areas, the development of the tourism and leisure industry has achieved effective results in the process of poverty reduction in China and the world.

Tourism is one of the strongest drivers of the world economy and trade, contributing significantly to the economic improvement in developing countries. In response to the Sustainable Development Goals (SDGs), the United Nations World Tourism Organization (UNWTO) has formulated the Sustainable Tourism Poverty Eradication Plan, calling for the sustainable development of tourism to help people in poverty-stricken areas around the world get rid of poverty. The plan aims to achieve sustainable tourism by encouraging sustainable development in socioeconomic and ecological fields. UNWTO considers the plan to be an effective tool to make a tangible contribution to the SDGs, with tourism playing an important role in addressing extreme poverty, environmental sustainability, and global partnerships. This is especially true for developing African nations with limited economic opportunities in other sectors that often see tourism as an engine for economic and social development [20,21]. For example, tourism is Gambia's largest foreign exchange earner, accounting for about 16% of its GDP and one in seven jobs in the country [22]. China's targeted tourism poverty alleviation initiative has also achieved remarkable results, providing the Chinese example for the world's poverty reduction and the sustainable development of tourism. Zhao (2015) found that China's tourism development has a significant positive effect on economic growth, and the effect value should reach 0.1519, which is close to the conclusions arrived at by foreign scholars on China [23].

According to the UNWTO, the tourism sector now accounts for 10% of the world's GDP. It is an important part of the tertiary industry and also one of the fast-growing emerging industries in the world. More and more attention from policy-makers is being paid to tourism as an important tool for economic and social development, of which China is a prime example. Taleb Rifai, the former secretary general of the UNWTO, spoke highly of China's position in the world's tourism industry: "China has provided the best example for the development of the world's tourism industry—taking tourism as a priority area for development, fully unleashing its own potential on issues of rural development and poverty alleviation". Taleb Rifai believes that the Chinese model of tourism development and the measures China has taken on these issues can be used to guide the development of other destinations [24].

Among the economic forms related to the cultural and tourism industry, in addition to the key services provided by tourism, other industries also support poverty eradication directly, such as transportation, accommodation, food, beverage, handicrafts, and souvenirs. These tourism-related networks make services and products more diffuse, reaching beyond local areas [25]. These tourism-related services enable more local people to participate in the tourism industry, thus increasing local employment and income. Therefore, tourism also promotes the diversification of the local economy and industrial structure, especially in poor and developing areas where other economic options are limited [26]. As a multifaceted industry, the development of the cultural tourism industry will bring together many stakeholders to realize urban sustainable development.

### 2.4. Tourism Development and Environmental Effects

The relationship between tourism development, economic growth, and carbon dioxide emissions has been extensively studied. Tourism is recognized as an engine of local and global economic growth [27–29]. The economic benefits of tourism include, but are not limited to, increasing national tax revenue and improving employment. However, despite

its significant contribution to the economy, tourism is often blamed for its detrimental impact on environmental quality. Studies have found that greenhouse gas emissions resulting from current human practices exacerbate the adverse effects of climate change [30]. Recently, a study by Lenzen et al. (2018) reassessed global tourism-related emissions between 2009 and 2013 and found that carbon emissions from tourism have increased from 3.90 to 4.55 GtCO$_2$e, accounting for nearly 8.1% of global emissions [31]. According to a report (2008) by the United Nations World Tourism Organization (UNWTO), the transportation and accommodation sectors are the core contributors of global carbon emissions, accounting for 75% and 21%, respectively [32].

However, due to the intrinsic nature of tourism, behaviors such as transportation, accommodation, and tourism activities can affect environmental quality by increasing fossil fuel and energy consumption [33,34], and increasing carbon dioxide emissions [35], but at the same time, tourism can also provide ecological functions and services [36]. Dogru et al. studied the contradictory relationship between tourism development and environmental pollution in the context of the phenomenon of the environmental Kuznets curve, and found that tourism development in Italy, Luxembourg, and Slovakia had a significant positive impact on CO$_2$ emissions, while tourism development in Canada, the Czech Republic, and Turkey had a significant negative impact on CO$_2$ emissions, achieved by environmentally sustainable practices [27]. It was also found that with the development of tourism, the CO$_2$ emission caused by tourism has an asymmetric emission reduction effect through technological innovation [37].

Thus, we need to implement environmental regulation policies, develop people's attitudes towards environmental protection [38], promote energy efficiency and innovation, and use green technologies to prompt tourism development to have a positive impact on the environment [36,39].

## 3. Methodology

### 3.1. Study Area

Zhenjiang is located in the southwest region of the Jiangsu province in China, on the south bank of the lower reaches of the Yangtze River. It is one of the 27 cities in the central region of the Yangtze River Delta. There are three districts and three county-level cities under Zhenjiang's jurisdiction, measuring 95.5 km wide from east to west, 76.9 km long from north to south, totaling 3840 square kilometers. Zhenjiang is the center of the north wing of the Yangtze River Delta, the core city of the Nanjing metropolitan area, and an important part of the national modernization demonstration area in southern Jiangsu. In 2012, Zhenjiang was listed as the second batch of national low-carbon pilot cities by the NDRC. Zhenjiang has hosted the "International Low-carbon (Zhenjiang) Conference" four times consecutively, from 2016 to 2019. During the 13th Five-Year Plan period, Zhenjiang's energy consumption per unit GDP decreased by 14.9%. Ecological economy is the exploration direction of a "low-carbon Zhenjiang".

As a green and low-carbon industry, the tourism industry is an important representation of the city's sustainable development capacity. Tourism accounts for an increasing proportion of Zhenjiang's industrial GDP year-by-year. The total tourism revenue in 2019 was 102 billion Yuan, accounting for 25% of its GDP. Zhenjiang is known as the "urban forest". With the deepening of low-carbon city construction, a low-carbon tourism system has been gradually formed. The Nanshan Scenic area of "Three Mountains" is the core of cultural tourism, the Yangtze River wetland and forest park protection promote the construction of green carbon sink, and the development of the Jinshan Bay tourism resort all lead the city's industrial transformation and upgrades.

At the same time, Zhenjiang's transportation industry also actively responds to the strategic layout of the tourism industry. Zhenjiang, with Nanjing in the west, Changzhou in the south, and Yangzhou in the north, is an important transportation center in East China. At the junction of Danyang city and Changzhou City is the Changzhou Benniu International Airport. Zhenjiang also has highways, railways, and other access to major

cities in China. In addition, Zhenjiang is the only junction between the Yangtze River and the Beijing–Hangzhou Grand Canal. As the third largest one-hundred-million-metric-ton port in the Yangtze River basin, the Zhenjiang Port can connect the river to the sea.

### 3.2. Data Sources

All data used in this study came from the China Tourism Statistical Yearbook, the China Energy Statistical Yearbook, and Zhenjiang Statistical Yearbooks from 2000–2019, as well as data provided by the Zhenjiang Statistics Bureau, Tourism Bureau, Ecology and Environment Bureau, and other government departments [3,40,41]. The data from the statistical yearbook included the total income of tourism; GDP; the output value of the first, second, and tertiary industries; employment data of the tertiary industries; passenger turnover data of highway, railway, and civil aviation, and water transport involved in the carbon footprint of tourism transportation; as well as tourist hotel rooms and the average annual room occupancy rate involved in the carbon footprint of tourism accommodation. The data for carbon sink and tourism products came from government departments.

### 3.3. Analysis

3.3.1. Carbon Storage

According to literature research and the research results of Johnson, W.C. et al., K. et al., Fang J. et al., and Zhao, M. et al. [42–45], the carbon storage of Zhenjiang City is divided into four parts: arbor forest, bamboo forest, economic forest, and shrub forest. The calculation formula of carbon storage in the arbor forest is shown as follows:

$$C_{arbor} = V_{arbor} * SVD_{arbor} * BEF_{arbor} * CF \tag{1}$$

where $C_{arbor}$ represents carbon storage of the arbor forest (t), $V_{arbor}$ represents the arbor forest stock volume per unit area (m$^3$/ha), $SVD_{arbor}$ represents basic wood density (t/m$^3$), $BEF_{arbor}$ represents the biomass expansion factor, and $CF$ represents carbon fraction.

The calculation formula of carbon storage in the bamboo forest, economic forest, and shrub forest is shown as follows:

$$C_i = V_i * SVD_i * CF \tag{2}$$

where $C_i$ represents forest carbon storage (t); $V_i$ represents forest stock volume per unit area (m$^3$/ha); $SVD_i$ represents basic wood density (t/m$^3$); $CF$ represents carbon fraction; and $i$ = the bamboo forest, economic forest, and shrub forest.

3.3.2. Annual Absorption of Carbon Sink

The calculation formula of annual absorption of forest carbon sink is shown as follows:

$$\Delta C_{biomass} = \Delta C_{arbor} + \Delta C_{scattered/adjacent/sparse} + \Delta C_{bamboo/economic/shrub} - \Delta C_{consumption} \tag{3}$$

where $\Delta C_{biomass}$ represents carbon storage change of forest and other woody biomass (t); $\Delta C_{arbor}$ represents carbon uptake of the arbor forest (t); $\Delta C_{scattered/adjacent/sparse}$ represents carbon uptake of scattered trees, adjacent trees, and sparse trees; $\Delta C_{bamboo/economic/shrub}$ represents carbon storage change of the bamboo forest, economic forest, and shrub forest; and $\Delta C_{consumption}$ represents carbon emissions from living wood consumption [42–45]. All data processing and chart making based on the collected data in this paper were completed in Excel and ArcGIS 10.2.

3.3.3. Estimation of Tourism Carbon Footprint

According to literature research, the carbon footprint of tourism mainly comes from tourism transportation, accommodation, and tourism activities. The calculation of the

carbon footprint of each part mainly refers to the research results of Shi et al. [46]. The calculation formula is as follows:

$$C = C_T + C_A + C_O \tag{4}$$

where $C$ represents the total tourism-related $CO_2$ emissions; and $C_T$, $C_A$, and $C_O$ represent the $CO_2$ emissions from transportation, accommodation, and tourism activities, respectively. The calculation formula for each part is as follows:

$$C_T = \sum_{i=1}^{4} T_i * \alpha_i \tag{5}$$

where $i$ represents the mode of transportation, including road, railway, air, and water transportation; $T_i$ represents the passenger turnover of type $i$ (in units of pkm); and $\alpha_i$ represents the $CO_2$ emission coefficient of type $i$ (in units of g $CO_2$/pkm). Referring to the research results of Shi Peihua [46], Wu Wenhua [47], and Kuo et al. [48], the $CO_2$ emission coefficients of road, railway, air, and water transportation are 133 g $CO_2$/pkm, 27 g $CO_2$/pkm, 137 g $CO_2$/pkm, and 106 g $CO_2$/pkm, respectively.

$$C_A = B * R * \beta * d \tag{6}$$

where $B$ represents the number of beds in tourism hotels; $R$ represents the average annual bed occupancy rate; $\beta$ represents the $CO_2$ emission coefficient of each bed per night (in units of g$CO_2$/ night/bed); and $d$ is equal to 365 days, the number of days in a year.

$$C_O = \sum_{k=1}^{5} N_k * \gamma_k \tag{7}$$

where, $k$ represents the types of tourism activities, including sightseeing, leisure, business, visiting relatives and friends, and others; $N_k$ represents the number of tourists in type $k$ tourism activities; and $\gamma_k$ represents the $CO_2$ emission coefficient of type $k$ activities (in units of g$CO_2$/per capita).

## 4. Results

According to the "Preliminary Implementation Plan of Pilot Low-carbon City in Zhenjiang" issued in 2012, the overall goals of low-carbon city construction in Zhenjiang are as follows: (1) improving the quality of economic development; (2) optimizing industrial structure and energy structure; and (3) reducing carbon emission intensity to build Zhenjiang into a modern landscape garden city with distinctive features and an elegant quality. With the landscape garden city goals, Zhenjiang tries to promote the layout of the modern tourism industry in two directions: constructing a modern industrial system through transformation, and promoting a green strategy, such as increasing carbon sink.

### 4.1. Overview of the Study Area

Zhenjiang has a long history and culture. From 1928 to 1949, Zhenjiang was the capital of the Jiangsu province. Located at the intersection of the Yangtze River and the Beijing–Hangzhou Grand Canal, Zhenjiang is the coordinate of the "River Overpass" in China. Zhenjiang has rich tourism resources such as a river island, city forest, landscapes, religion, a countryside, and food, forming a rich and diverse tourism product system including cultural leisure, landscape sightseeing, rural leisure, and vacationing to improve health. As shown in Figure 1, the distribution of the ecological tourism park in Zhenjiang is consistent with the distribution of forest on the land-use map.

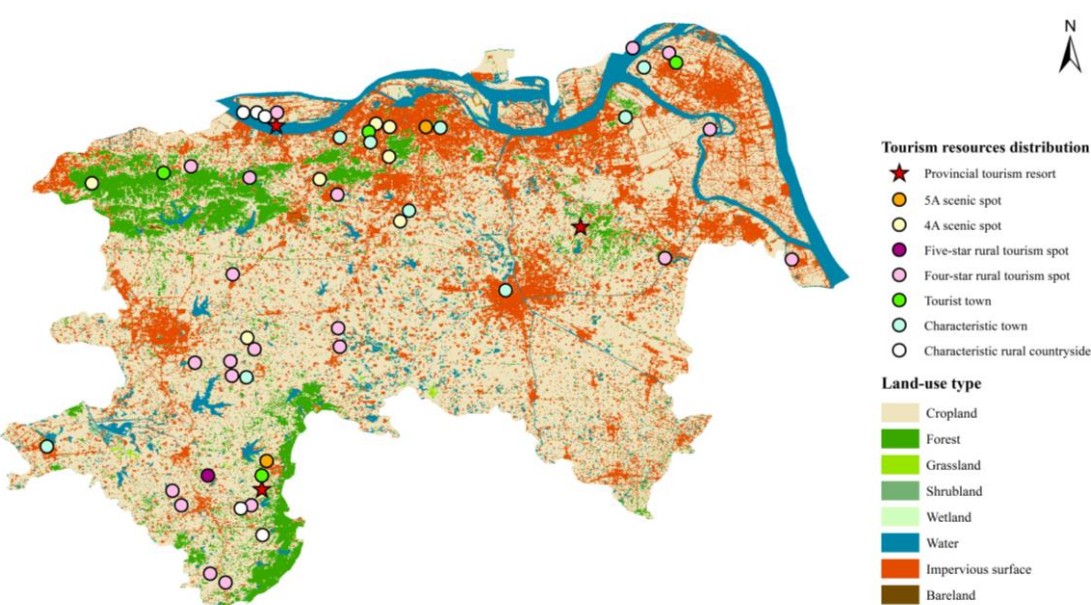

**Figure 1.** Land-use and tourism distribution map of Zhenjiang City.

As of 2019, Zhenjiang has 35 A-level scenic spots, including two 5A-level scenic spots, seven 4A-level scenic spots, and twelve 3A-level scenic spots. There are three provincial tourist resorts and 86 provincial star rural tourist areas. There are 25 star-hotels, including three 5-star hotels. It has 120 travel agencies, including 35 star-travel agencies. It received 71 million tourists from home and abroad, which increased by 8.8% over the previous year. The total tourism revenue reached 102 billion Yuan, an increase of 9.6%.

### 4.2. The Proportion of CTI in GDP

The proportion of tourism in Zhenjiang's industrial GDP is increasing year-by-year (Figure 2). In 2019, the total tourism revenue was 102 billion Yuan, accounting for 25% of GDP, and increased by more than 9% from 2017 to 2019.

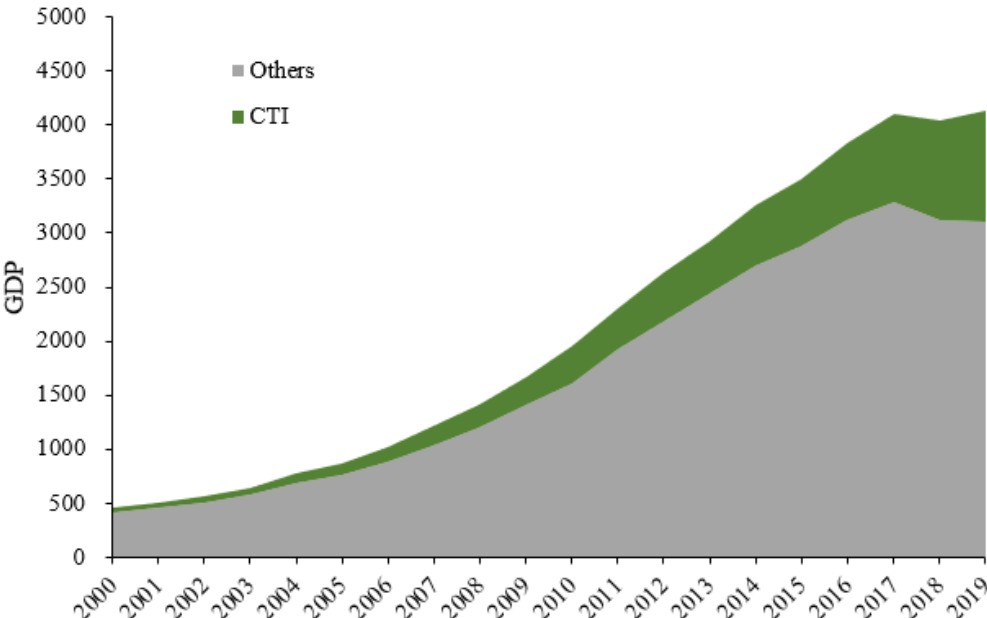

**Figure 2.** The proportion of CTI in GDP (unit: 100 million Yuan).

### 4.3. Contribution of CTI Development to Carbon Sink

The forest carbon sink in Zhenjiang mainly comes from land-use change and forestry. As shown in Table 1, the carbon storage of forest carbon sink in Zhenjiang is as high as 3.72 million Mt of carbon dioxide by calculating the carbon storage of the arbor forest, bamboo forest, economic forest, and shrub forest. The results show that the forest in Zhenjiang can absorb about 150,000 Mt of $CO_2$ every year.

**Table 1.** Estimation of carbon storage in Zhenjiang.

| Forest Type | Carbon Storage ($10^3$ Mt) | Total Carbon Storage ($10^3$ Mt) | Annual Carbon Sink ($10^3$ Mt) |
|---|---|---|---|
| Arbor Forest | 2190 | | |
| Bamboo Forest | 410 | | |
| Economic Forest | 860 | 3720 | 150 |
| Shrub Forest | 260 | | |

### 4.4. Visitation and Carbon Footprint Generated by CTI

The CTI-related carbon footprint in Zhenjiang increased from 509,800 Mt $CO_2$ in 2009 to 699,800 Mt $CO_2$ in 2012, and then gradually decreased and stabilized. In 2019, the total carbon footprint was 479,900 Mt $CO_2$, among which transportation, accommodation, and tourism activities accounted for 86.2%, 3.7%, and 10.1%. Compared to the previous data, transportation is the most significant contributor in the estimation of the tourism carbon footprint, which is consistent with the conclusions of the World Tourism Organization—global tourism transportation carbon emissions account for 90% of total tourism carbon emissions (UNWTO, 2009).

As shown in Figure 3 and Table 2, in addition to the overall increase in the total carbon footprint of tourism activities, the total carbon footprint of tourism transportation and accommodation showed a trend of first rising and then falling from 2009 to 2019. Among them, highways accounted for the largest in the transportation carbon footprint, but the proportion is gradually decreasing. With the development of China's civil aviation and railway industry, the carbon footprints of the two have increased yearly. As can be seen in Figure 3b, the carbon footprint of tourism accommodation has been greatly reduced after 2013, owing to green and low-carbon hotel management. As shown in Figure 3c, tourism activities, mainly composed of sightseeing, leisure, business and visiting friends, and other activities, demonstrate a rising trend year-by-year. Among them, visiting and sightseeing make up the main part of the carbon footprint, accounting for 42.6% and 26.0%, respectively.

**Table 2.** Visitation and carbon footprint generated by CTI in Zhenjiang.

| Year | Visitation ($10^3$ Persons) | Carbon Footprint ($10^3$ tons $CO_2$) | | | | Carbon Footprint per Capita (kg $CO_2$) |
|---|---|---|---|---|---|---|
| | | Transportation | Accommodation | Other Activities | Total | |
| 2009 | 22,420 | 468.4 (91.9%) | 26.1 (5.1%) | 15.3 (3.0%) | 509.8 | 22.7 |
| 2010 | 26,070 | 527.4 (90.4%) | 38.1 (6.5%) | 17.7 (3.0%) | 583.3 | 22.4 |
| 2011 | 31,000 | 582.2 (90.7%) | 38.5 (6.0%) | 21.1 (3.3%) | 641.8 | 20.7 |
| 2012 | 35,020 | 635.5 (90.8%) | 40.4 (5.8%) | 23.8 (3.4%) | 699.8 | 20.0 |
| 2013 | 38,950 | 418 (86.5%) | 38.6 (8.0%) | 26.5 (5.5%) | 483.2 | 12.4 |
| 2014 | 43,850 | 431.3 (88.2%) | 27.6 (5.6%) | 29.9 (6.1%) | 488.8 | 11.1 |
| 2015 | 48,030 | 435.6 (89.3%) | 19.5 (4.0%) | 32.7 (6.7%) | 487.7 | 10.2 |
| 2016 | 53,480 | 435.4 (88.3%) | 21.2 (4.3%) | 36.4 (7.4%) | 493.0 | 9.2 |
| 2017 | 59,650 | 426.1 (87.5%) | 20.5 (4.2%) | 40.6 (8.3%) | 487.2 | 8.2 |
| 2018 | 65,470 | 413.2 (86.0%) | 22.6 (4.7%) | 44.6 (9.3%) | 480.3 | 7.3 |
| 2019 | 71,210 | 413.6 (86.2%) | 17.8 (3.7%) | 48.5 (10.1%) | 479.9 | 6.7 |

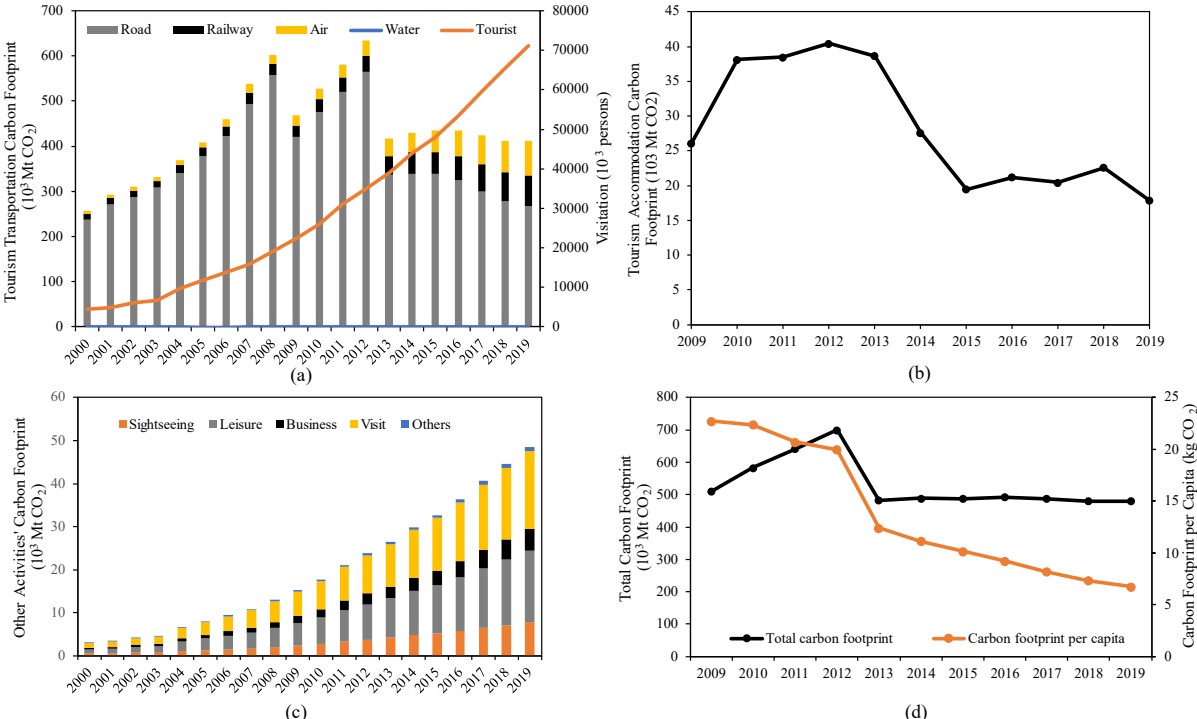

**Figure 3.** The relationship between visitation and carbon footprint generated by CTI. (**a**) The visitation and carbon footprint generated by tourism transportation in Zhenjiang. (**b**) Carbon footprint generated by tourism accommodation in Zhenjiang. (**c**) Carbon footprint generated by other tourism activities in Zhenjiang. (**d**) The visitation and total carbon footprint generated by CTI.

Thanks to the gradually forming conception of low-carbon tourism, the carbon footprint per capita has presented a downward trend on the whole, which is also the result of low-carbon promotion by the Zhenjiang municipal government and enterprises.

### 4.5. CTI-Promoted Industrial Transformation

The cultural and tourism industry plays an important role in promoting the transformation of the traditional industrial structure. The development of the cultural and tourism industry can adjust the relationship between the secondary industry and the tertiary industry, but can also derive new categories from the traditional service industry and is conducive to upgrading the traditional service industry. As shown in Figure 4, with the development of CTI, the tertiary industry occupies an increasing proportion. Therefore, CTI development plays an obvious role in promoting the transformation of the traditional industrial structure dominated by the primary and secondary industries in Zhenjiang.

### 4.6. Increase in CTI-Related Employment

The reason why tourism can not only develop itself, but also promote the sustainable development of the city is that, in addition to its obvious environmental friendliness and its ability to drive the transformation and upgrades of the first, second, and tertiary industries, CTI also has a strong driving force for employment. CTI is one of the industries that can consider economic growth and employment, and one of the industries with the most obvious effect of enriching people.

As seen in Figure 5, with the continuous development of CTI and the rise and expansion of the tertiary industry, the employment related to the development of CTI in Zhenjiang increased significantly, which can further drive poverty eradication and promote sustainable development of the city.

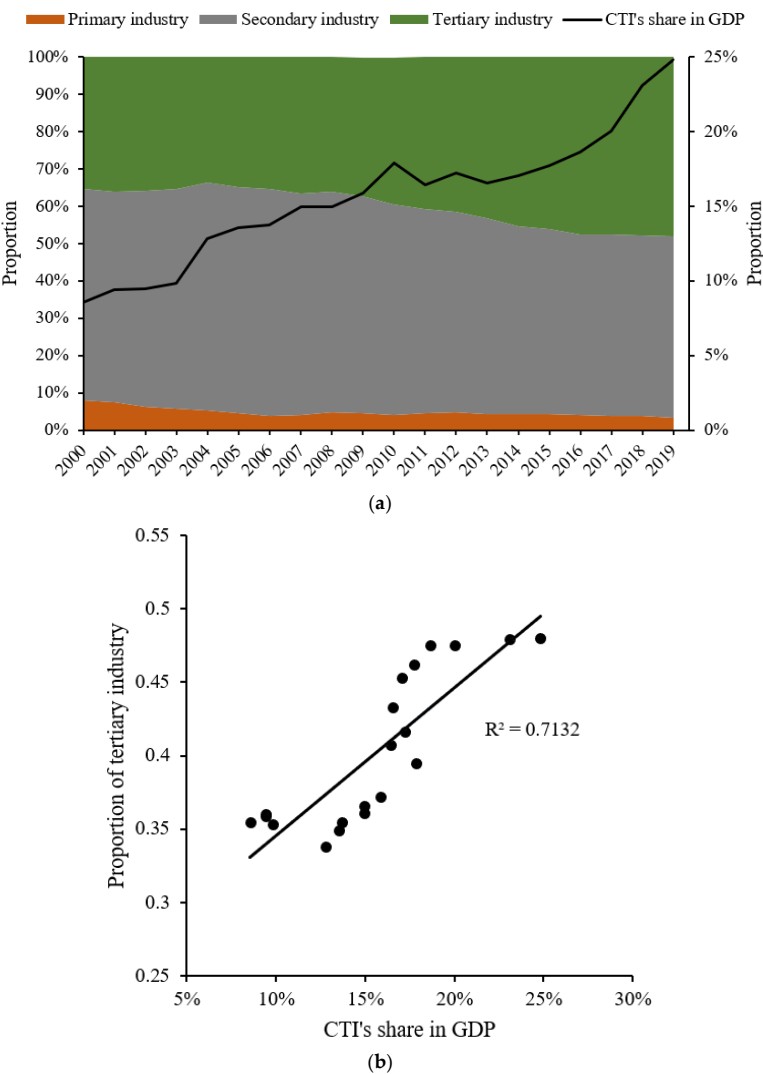

**Figure 4.** CTI promotes industrial transformation. (**a**) The proportion of primary, secondary, and tertiary industries in Zhenjiang. (**b**) The relationship between CTI and tertiary industry.

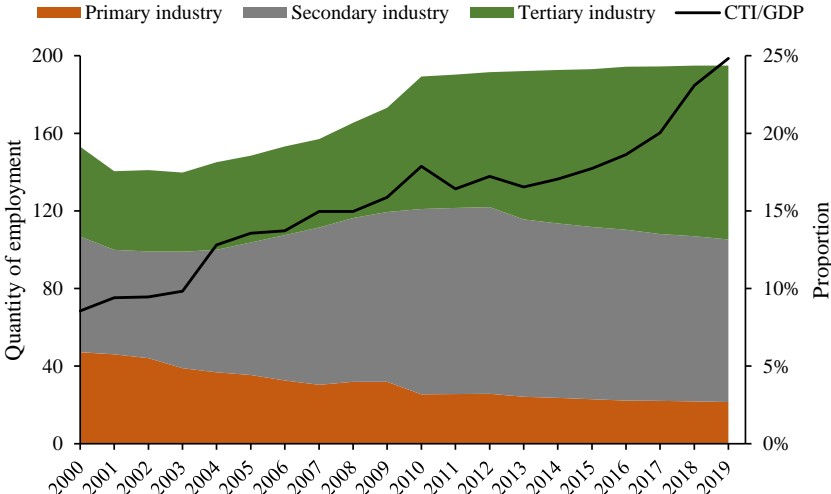

**Figure 5.** The proportion of employment in the primary, secondary, and tertiary industries in Zhenjiang (unit: 10 thousand people).

*4.7. Analysis in Tourism-Generating Region of Tourism Products*

The excellent tourism foundation of Zhenjiang, especially the convenient transportation conditions, guarantees the accessibility of scenic spots, which have greatly expanded the radiation range of the tourism-generating region. The five-pointed red star represents the geographical location of Zhenjiang while the size of the blue circle in Figure 6 represents the percentage of purchases of tourism products from different regions. The larger the circle is, the more purchases were made. As seen in Figure 6, the top three cities are Nanjing, Shanghai, and Beijing. In addition to the distance factor, both Shanghai's and Beijing's high percentages are due to the convenient transportation conditions. For example, it only takes 1–2 h to get from Shanghai to Zhenjiang by high-speed rail, and the Runyang Yangtze River Bridge connects to the Beijing–Shanghai Expressway, allowing commuters to reach Beijing directly.

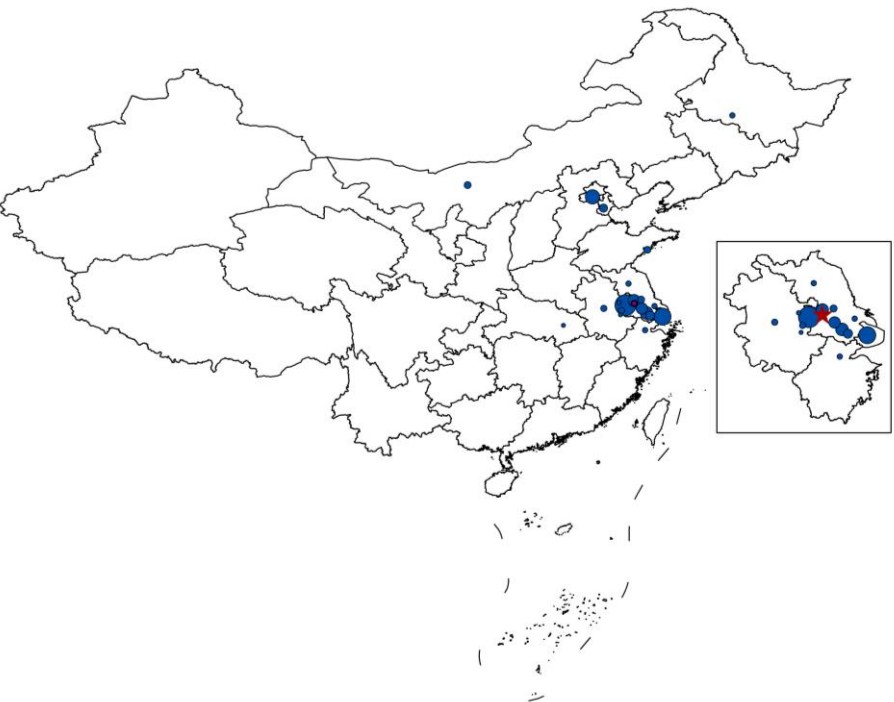

**Figure 6.** Distribution of tourists visiting Zhenjiang.

## 5. Conclusions and Discussion

*5.1. Conclusions and Suggestions*

5.1.1. Advocate Green and Low-Carbon Travel Modes

According to the analysis of the tourism-generated carbon footprint in Zhenjiang, transportation is the main source of the city's carbon footprint, accounting for 86.2% in 2019, which is consistent with the results of a report by the United Nations World Tourism Organization [32]. Therefore, the main direction of energy-saving and emission-reducing tourism in Zhenjiang should also be to improve the use of tourism vehicles. In transportation carbon emissions, roads account for 64.5%. With the development of the social economy, self-driving travel has become the first choice for people in suburban tourism, and the resulting pressure on carbon emissions from private car travel is greater. Therefore, relying on urban public transportation planning, the construction of a reasonable urban tourism transportation system can effectively reduce transportation carbon emissions and promote low-carbon travel modes. Secondly, the government can vigorously promote the use of new energy vehicles due to their relatively low carbon dioxide emissions. By implementing measures such as tax exemptions for car purchases and easy access to license plates, the popularization of new energy vehicles can be encouraged, thus making self-driving travel enter the era of low-carbon emissions.

### 5.1.2. Develop the Concept of "All-for-One" Tourism

As far as the CTI in Zhenjiang is concerned, the tourist attractions are too scattered and there are still some deficiencies, which weaken the advantages of tourism in Zhenjiang. Therefore, under the leadership of the government, Zhenjiang has carried out reasonable integration and replanning of tourist attractions [49]. With the proposal of the concept of "all-for-one" tourism, the Zhenjiang municipal government also issued the "Outline of All-inclusive Tourism Planning of Zhenjiang" in 2019. "All-for-one" tourism is a new planning concept proposed by Chinese tourism management practitioners that is expected to guide the transformation and upgrade of the country's regional tourism industry. It considers tourism to be the dominant industry, integrating economic and social resources, ecological environment policies and regulations, and other resources in the region in order to achieve integrated industrial development. The outline identifies tourism as the first pillar industry of Zhenjiang's modern service industry. Zhenjiang is committed to playing the role of tourism as an engine of a green and happy industry, and building Zhenjiang into a tourism destination featuring landscapes, culture, leisure, and vacationing to improve health. Additionally, Zhenjiang has also set the development goals of annual tourism growth at 7.2%, annual tourism growth at 5.5%, and an overall good air quality rate at more than 66.5%.

This multi-dimensional tourism management approach including ecological environment protection, cultural tourism resource protection, and overall environmental improvement can effectively ensure the overall protection of the natural ecosystem and promote biodiversity conservation and ecological environment restoration.

### 5.1.3. Improve the Attraction of Carbon Sink Tourism by Relying on Natural Conditions

As the important carbon sink for terrestrial ecosystems, forests are able to take carbon dioxide from the atmosphere, fixing it in vegetation or soil. The high forest coverage rate in Zhenjiang tourist scenic spots and the strong carbon sequestration capacity of the forest attach great importance to the absorption of carbon dioxide. In addition, carbon sink can also be used as a tourist attraction to show tourists the image of scenic spots. By combining with culture, it can effectively improve the tourist attraction of scenic spots, which is of great significance for low-carbon tourism and sustainable urban development. In view of the "all-for-one" tourism planning, Zhenjiang has guaranteed the protection of tourism carbon sink from the level of policies and regulations. In 2016, in order to strengthen the management of the Jinshan–Jiaoshan–Beigushan–Nanshan Scenic and Historic Interest Area, the conservation regulation of the Zhenjiang Sannanshan Scenic and Historic Interest Area was issued, which clearly stipulated the ecological protection area and protection plans. In 2020, the Zhenjiang municipal government issued the Zhenjiang Ancient Canal Protection Plan for the ancient canal located at the starting point of the Jiangnan section of the original Beijing–Hangzhou Grand Canal, effectively guaranteeing the implementation of river management and protection. The tourism resources guarantee that the system established in Zhenjiang will provide strong support for ecological protection and examples for urban tourism management.

The transformation of the urban development model is a profound revolution, and the sustainable development city led by the cultural and tourism industry should not only transform towards the goal of becoming a city of tourism, but also move towards the goal of becoming low carbon. As the sunrise industry in the 21st century, tourism meets the strategic requirements of promoting sustainable economic development, facilitating employment, and improving the ecological environment in the industrial transformation of resource-based cities. Therefore, vigorously developing tourism has become the ideal choice for resource-based cities to implement a positive industrial transformation strategy. In the face of China's current national strategy to achieve carbon peak by 2030 and carbon neutral by 2060, we should also seize the opportunity to seek a sustainable development path in accordance with the characteristics of different cities. Zhenjiang's model of the

cultural and tourism industry will also provide an example for other cities in China and the world.

*5.2. Limitations and Future Perspectives*

The description of the current situation and total amount of the tourism carbon footprint in this paper is only a preliminary estimate based on existing data. Due to the complexity of the CTI system, when calculating the tourism-related $CO_2$ emissions, in addition to the direct processes such as transportation, accommodation, and tourism activities, indirect processes should also be included, such as the operation and management of tourist attractions and solid waste. In addition, due to the limitation of statistical data, it is not possible to cover all sectors in the estimation of carbon footprints in this paper. For example, the data of star-rated hotels are used to represent the carbon emissions of the entire accommodation industry. In addition, the core parameters, such as the $\alpha$, $\beta$, and $\gamma$ in Formulas (5)–(7), are the key to the accuracy of the estimation results. However, limited by the research data of previous years, we are temporarily unable to obtain accurate local survey data. Therefore, this paper estimates each part of the tourism industry based on the empirical data of the existing surveys and research. In fact, with the vigorous development and technological innovation of low-carbon tourism in Zhenjiang, the energy-saving and emission-reducing technologies in all aspects have been improved, so the unit emission value should decrease. However, this will not affect the results and cause a magnitude of errors, so overall, the estimation results in this paper are credible for understanding the carbon footprint of CTI in Zhenjiang.

As the scale of tourism continues to grow, the impact of tourism on climate and the environment has attracted increasing attention from all walks of life. The main body of tourism's impact on the environment essentially stems from human activities, which require the depth and breadth of research work. Therefore, in order to improve the understanding of tourism environmental issues in a more comprehensive way in the future, the relevant departments of various countries are required to pay attention to them, and further improvements are needed in the acquisition of basic data and research methods.

**Author Contributions:** Conceptualization, R.L.; Data curation, R.L. and Z.Q.; Methodology, R.L. and Z.Q.; Resources, R.L.; Validation, Z.Q.; Visualization, Z.Q.; Writing–original draft, Z.Q.; Writing–review & editing, Z.Q. All authors have read and agreed to the published version of the manuscript.

**Funding:** This research received no external funding.

**Data Availability Statement:** The data presented in this study are available on request from the corresponding author.

**Conflicts of Interest:** The authors declare no conflict of interest.

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
