# Peer review of "Urban Sustainable Development Empowered by Cultural and Tourism Industries: Using Zhenjiang as an Example"

_sustainability, doi:10.3390/su141912884_

Round 1

Reviewer 1 Report

Thank you for the opportunity to review your paper. The paper takes Zhenjiang City in East China as an example and argues that its industrial culture and tourism industry (CTI) contributes to the sustainable development of the metropolitan area as an important low-carbon industry. There are some suggestions for optimizing this paper as follows:

1. The Sustainability journal expects papers to make a significant theoretical and/or methodological contribution to the sustainable tourism literature. While the findings of this paper provide useful practical implications for case studies, theoretical or methodological contributions to sustainability in tourism should be even more important.

2. The literature review is not sufficiently focused on the topic, and a summary of the literature review is needed to lead to the study of this paper.

3. How did the formulae for Carbon Storage and Annual Absorption of Carbon Sink come about? Why should this formula be chosen? Please explain or cite it in the text.

4. The first paragraph of the conclusion tells the overall goal of the construction of a low-carbon city in Zhenjiang, which belongs to the policy content, and I don't understand why the author put it in the conclusion part, which seems to be more suitable for the introduction part.

5.The section "Overview of Zhenjiang Tourism" provides an overview of the development of tourism in Zhenjiang, which should be more of an overview of the study area, so please adjust it appropriately.

6. The paper mentions that Zhenjiang's low-carbon sustainable development model will provide successful experiences for other cities in China and abroad. What experience has been provided? Please summarize them instead of just listing the relevant policies.

7 Add a sub-section on the limitations of the study and future perspectives.

Reviewer 2 Report

Dear authors, The article submitted for review contains the results of an interesting study that is valuable because of the impact that cultural and tourism industries have on urban areas. To improve the quality of the article, the authors should consider supplementing the presentation of the results (analysis) with some quantitative information, as well as showing the link between the visitation rate and the carbon footprint generated by the cultural and tourism industry.

Reviewer 3 Report

Authors must also acknowledge and discuss the negative implications stressed by some studies, which claim CO2 emissions are positively correlated with the number of arrivals/visitors. The CO2 emissions of the transportation industry in mass tourism context must be mitigated and/or compensated. Please update your literature review with the following contributions 

Dogru, T., Bulut, U., Kocak, E., Isik, C., Suess, C., & Sirakaya-Turk, E. (2020). The nexus between tourism, economic growth, renewable energy consumption, and carbon dioxide emissions: contemporary evidence from OECD countries. Environmental Science and Pollution Research27(32), 40930-40948.

Balsalobre-Lorente, D., & Leitão, N. C. (2020). The role of tourism, trade, renewable energy use and carbon dioxide emissions on economic growth: evidence of tourism-led growth hypothesis in EU-28. Environmental Science and Pollution Research27(36), 45883-45896.

Razzaq, A., Sharif, A., Ahmad, P., & Jermsittiparsert, K. (2021). Asymmetric role of tourism development and technology innovation on carbon dioxide emission reduction in the Chinese economy: Fresh insights from QARDL approach. Sustainable Development29(1), 176-193.

Round 2

Reviewer 1 Report

All comments addressed by the authors.